# P+: Extended Textual Conditioning in Text-to-Image Generation

## Abstract

We introduce an Extended Textual Conditioning space in text-to-image diffusion models, referred to as $\mathcal{P}+$. This space consists of multiple textual conditions, derived from per-layer prompts, each corresponding to a cross-attention layer of the denoising U-net of the diffusion model. We show that the extended space provides greater control over the synthesis process. We further introduce Extended Textual Inversion (XTI), which inverts concepts into $\mathcal{P}+$, such that they are represented with per-layer tokens. We show that XTI is more expressive and precise, and converges faster than the original Textual Inversion (TI) space. Compared to baselines, XTI achieves much better reconstruction and editability without the need to balance these two goals. We conduct a series of extensive experiments to analyze and understand the properties of the new space, and to showcase the effectiveness of our method for personalizing text-to-image models. Furthermore, we utilize the unique properties of this space to achieve previously unattainable results in object-style mixing using text-to-image models.

## 1 Introduction

Neural generative models have advanced the field of image synthesis, allowing us to create incredibly expressive and diverse images. Yet, recent breakthroughs in text-to-image models based on large language-image models have taken this field to new heights and stunned us with their ability to generate images from textual descriptions, providing a powerful tool for creative expression, visualization, and design.

Recent Text-to-Image models typically use a revers diffusion process to generate images from noisy tensors, performed with a U-Net denoiser model. The network uses multiple cross-attention layers, at different resolutions, to inject information from a conditioning text prompt. Figure 1 (left) shows the common text-conditioning flow: the textual prompt embedding $p$, element of the space that we denote as $\mathcal{P}$, is passed to multiple cross-attention layers of the denoising U-net model. In this paper, we introduce the *Extended Textual Conditioning* space. This space, referred to as $\mathcal{P}+$ space, consists of $n$ textual conditions $\{p_1, p_2, ...p_n\}$, where each $p_i$ is injected to the corresponding $i$-th cross-attention layer in the U-net (see Figure 1 (right)). We show that $\mathcal{P}+$ space is more expressive, disentangled, and provides better control on the synthesized image. As will be analyzed in this paper, different layers tend to control different aspects of the synthesized image. In particular, the coarse layers primarily affect the structure of the image, while the fine layers predominantly influence its appearance.

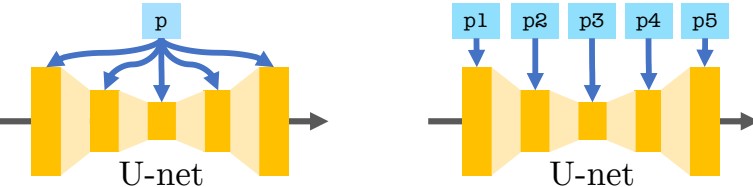

Figure 1: $\mathcal{P}$ **vs.** $\mathcal{P}+$. Standard textual conditioning, where a single text embedding is injected to the network (left), vs. our proposed extended conditioning, where different embeddings are injected into different layers of the U-net (right).

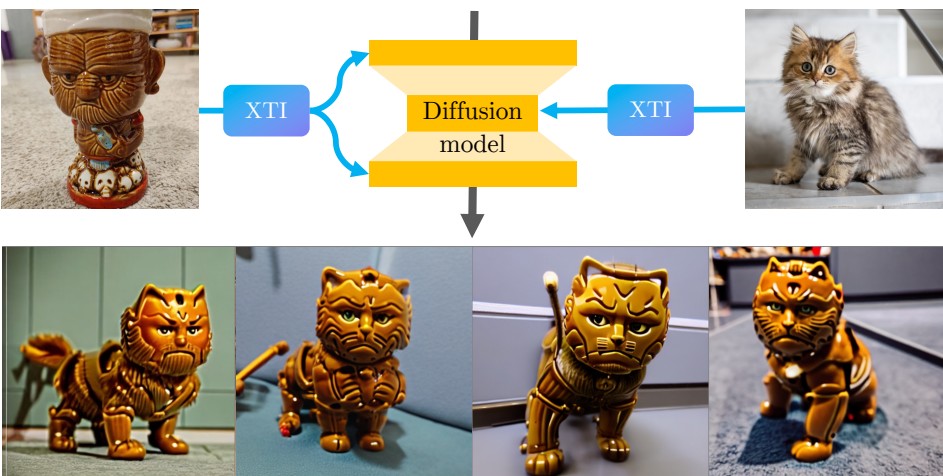

Figure 2: **Shape-Style Mixing in XTI.** The extended textual space allows subjects mixing conducted by two separate extended textual inversions (XTIs). The inversion of the kitten (right) is injected to the coarse inner layers of the U-net, affecting the shape of the generated image, and the inversion of the cup (left) is injected to the outer layers, affecting the style and appearance.

Our Extended Textual Conditioning space paves the way to a particularly exciting advancement in the domain of personalization of text-to-image models Gal et al. (2022); Ruiz et al. (2022), where the model learns to reproduce a specific subject depicted on a few input images, in different contexts. This inversion process results in a new conditioning token that represents the subject. Then it can be employed in a text prompt to produce diverse and novel images where the subject appears in a new context.

To this end, we introduce Extended Textual Inversion (XTI), where a subject portrayed by a few images is represented as a set of token embeddings, one per layer. Our findings reveal that Our results indicate that the optimized embeddings in XTI not only converge faster compared to those in the textual inversion baseline, but they also enhance the reconstruction quality without sacrificing the ability to edit.

Furthermore, we leverage the distinctive characteristics of $\mathcal{P}+$ to advance the state-of-the-art object-appearance mixing through text-to-image generation. Specifically, we employ the insertion of inverted tokens of diverse subjects into the different layers to capitalize on the inherent shape-style disentanglement exhibited by these layers. This approach enables us to achieve previously unattainable results as shown in Figure 2.

In summary, the contributions of our paper are:

1. We introduce $\mathcal{P}+$, the Extended Textual Conditioning space, which is represented by a per-layer token embedding. $\mathcal{P}+$ is more expressive and disentangled, allowing for better control over different aspects of the synthesized image's structure and appearance.

2. We propose the Extended Textual Inversion (XTI) method to represent a subject in $\mathcal{P}+$ using a set of token embeddings, improving convergence speed and reconstruction quality (compared to Textual Inversion) without sacrificing editability.

3. We demonstrate previously unattainable results of object-appearance mixing through text-to-image generation.

## 2 RELATED WORKS

### 2.1 EXTENDED SPACES IN GENERATIVE MODELS

Exploring neural sub-spaces in generative models has been extensively explored, most notably in StyleGAN Karras et al. (2020; 2019). The extended textual conditioning $\mathcal{P}+$ is reminiscent of StyleGAN's extended latent space Abdal et al. (2019; 2020), also commonly referred to as $\mathcal{W}+$. Similar to $\mathcal{W}+$, $\mathcal{P}+$ is significantly more expressive, where instead of a single code shared by all layers, there is one per layer. However, while $\mathcal{W}+$ is an extended latent space, here the extended

space relates to the textual conditions used by the network. It should be noted, though, that while $\mathcal{W}+$ is expressive, the extended code is less editable Tov et al. (2021). In contrast, $\mathcal{P}+$ remains practically as editable as $\mathcal{P}$. In addition, other sub-spaces lay within deeper and more disentangled layers Wu et al. (2021) have been explored and exploited in various editing and synthesis applications Bermano et al. (2022).

In the case of text-to-image diffusion models, the denoising U-net, which is the core model of most of the text-to-image diffusion models, is usually conditioned by text prompts via a set of cross-attention layers Ramesh et al. (2022); Rombach et al. (2021); Saharia et al. (2022). In many neural architectures, different layers are responsible for different abstraction levels Bau et al. (2020); Karras et al. (2019); Voynov & Babenko (2020); Zeiler & Fergus (2014); Ghiasi et al. (2022). It is natural to anticipate that the diffusion denoising U-Net backbone operates in a similar manner, with different textual descriptions and attributes proving beneficial at different layers.

## 2.2 TEXT-DRIVEN EDITING

Recently there has been a significant advancement in generating images based on textual inputs Chang et al. (2023); Ramesh et al. (2022); Rombach et al. (2021); Saharia et al. (2022), where most of them exploit the powerful architecture of diffusion models Ho et al. (2020); Rombach et al. (2021); Sohl-Dickstein et al. (2015); Song et al. (2020); Song & Ermon (2019).

In particular, recent works have attempted to adapt text-guided diffusion models to the fundamental problem of single-image editing, aiming to exploit their rich and diverse semantic knowledge of this generative prior. In a pioneering attempt, Meng et al. Meng et al. (2021) add noise to the input image and then perform a denoising process from a predefined step. Yet, they struggle to accurately preserve the input image details, which were preserved by a user provided mask in other works Avrahami et al. (2022b;a); Nichol et al. (2021). DiffEdit Couairon et al. (2022) employs DDIM inversion for image editing, but to prevent any resulting distortion, it generates a mask automatically that allows background preservation.

Text-only editing approaches split into approach that supports global editing Crowson et al. (2022); Kim & Ye (2021); Kwon & Ye (2021); Patashnik et al. (2021); Liew et al. (2022), and local editing Bar-Tal et al. (2022); Wang et al. (2022). Prompt-to-prompt Hertz et al. (2022) introduces an intuitive editing technique that enables the manipulation of local or global details by injecting internal cross-attention maps. To allow prompt-to-prompt to be applied to real images, Null-Text Inversion Mokady et al. (2022) is proposed as means to invert real images into the latent space of the diffusion model. Imagic Kawar et al. (2022a) and UniTune Valevski et al. (2022) have demonstrated impressive text-driven editing capabilities, but require the costly fine-tuning of the model. Instruct-Pix2Pix Brooks et al. (2023), Plug-and-Play Tumanyan et al. (2022), and pix2pix-zero Parmar et al. (2023) allow users to input an instruction or target prompt and manipulate real images accordingly.

## 2.3 PERSONALIZATION

Synthesizing particular concepts or subjects which are not widespread in the training data is a challenging task. This requires an *inversion* process that given input images would enable regenerating the depicted object using a generative model. Inversion has been studied extensively for GANs Bermano et al. (2022); Creswell & Bharath (2018); Lipton & Tripathi (2017); Xia et al. (2021); Yeh et al. (2017); Zhu et al. (2016), ranging from latent-based optimization Abdal et al. (2019; 2020) and encoders Richardson et al. (2020); Tov et al. (2021) to feature space encoders Wang et al. (2021) and fine-tuning of the model Alaluf et al. (2021); Roich et al. (2022); Nitzan et al. (2022).

The notion of personalization of text-to-image models has been shown to be a powerful technique. Personalization of models Kumari et al. (2023); Ruiz et al. (2022) in general, or of text tokens only Gal et al. (2022) has quickly been adapted for various applications Kawar et al. (2022b); Lin et al. (2022). In addition to their high computational cost, current methods face a clear trade-off between learning tokens that accurately capture concepts vs. avoidance of overfitting. This can result in learned tokens that are overly tuned to the input images, thus limiting their ability to generalize to new contexts or generate novel variations of the concept.

Similar to Textual Inversion (TI), our approach does not require any fine-tuning or modification of the weights, thus, reduces the risk of overfitting and degrading the editability capabilities. In contrast, our inversion process into $\mathcal{P}+$ is both faster and more precise, thanks to the greater number of tokens that improve reconstruction capabilities without sacrificing editability.

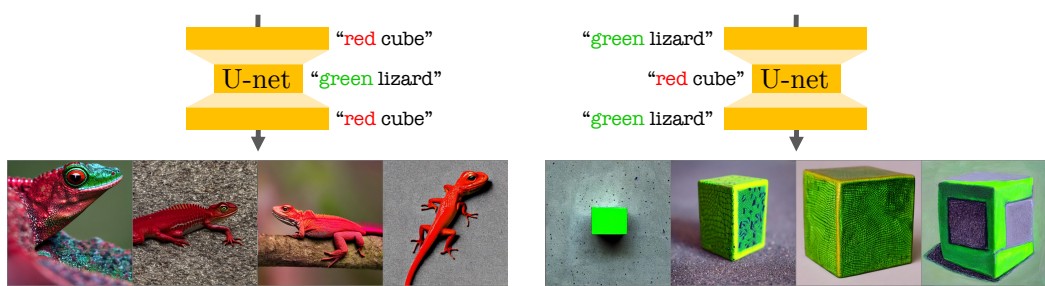

Figure 3: **Per-layer Prompting.** We provide different text prompts (a precursor to $\mathcal{P}+$) to different cross-attention layers in the denoising U-net. We see that color (`"red"`, `"green"`) is determined by the fine outer layers and content (`"cube"`, `"lizard"`) is determined by the coarse inner layers.

## 3 EXTENDED CONDITIONING SPACE

To engage the reader, we begin with a simple experiment on the publicly available Stable Diffusion model Rombach et al. (2022). We partitioned the cross-attention layers of the denoising U-net into two subsets: coarse layers with low spatial resolution and fine layers with high spatial resolution. We then used two conditioning prompts: `"red cube"` and `"green lizard"`, and injected one prompt into one subset of cross-attention layers, while injecting the second prompt into the other subset. The resulting generated images are provided in Figure 3. Notably, in the first run, the model generates a red lizard, by taking the subject from the coarse layers' text conditioning, and appearance from the fine layers' conditioning. Similarly, in the second run, it generates the green cube, once again taking the appearance from the fine layers and the subject from the coarse layers. This experiment suggests that the conditioning mechanism at different resolutions processes prompts differently, with different attributes exerting greater influence at different levels. With this in mind, our work aims to further explore this phenomenon and its potential applications.

In the following parts, we present the Extended Textual Conditioning space ($\mathcal{P}+$), outlining its principal attributes. We then introduce Extended Textual Inversion (XTI), demonstrating how $\mathcal{P}+$ can be leveraged to improve the trade-off between reconstruction and editability in the original Textual Inversion approach.

### 3.1 $\mathcal{P}+$ DEFINITION

Let $\mathcal{P}$ denote the *textual-conditioning space*. $\mathcal{P}$ refers to the space of token embeddings that are passed into the text encoder in a text-to-image diffusion model. To clarify the definition of this space, we provide a brief overview of the process that a given text prompt undergoes in the model before being injected into the denoising network.

Initially, the text tokenizer splits an input sentence into tokens, with a special token marking the end of the sentence (EOS). Each token corresponds to a pre-trained embedding that is retrieved from the embedding lookup table. Subsequently, these embeddings are concatenated and passed through a pre-trained text encoder, then injected to the cross-attention layers of the U-net model. In our work, we define $\mathcal{P}$ as the set of individual token embeddings that are passed to the text encoder. The process of injecting a text prompt into the network for a particular cross-attention layer is illustrated in Figure 4.

We next present the *Extended Textual Conditioning* space, denoted by $\mathcal{P}+$, which is defined as:

$$\mathcal{P}+ := \{p_1, p_2, ...p_n\},\tag{1}$$

where $p_i \in \mathcal{P}$ represents an individual token embedding corresponding to the $i$-th cross-attention layer in the denoising U-net. Figure 1 illustrates the conceptual difference between the two spaces, $\mathcal{P}$ (left) and $\mathcal{P}+$ (right).

With the definition of the new space, our diffusion model, previously conditioned on a single prompt, can now synthesize images conditioned on a series of prompts, each associated with the correspondent cross-attention layer.

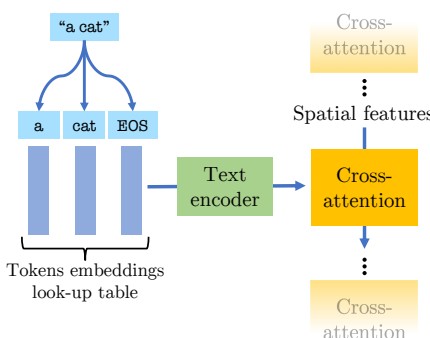

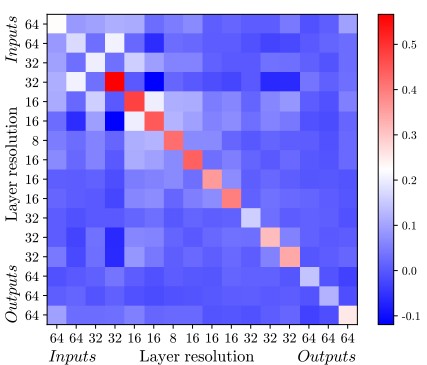

Figure 4: **Text-conditioning mechanism of a denoising diffusion model.** The prompt `"a cat"` is processed with a sentence tokenization by a pretrained textual encoder, and fed into a cross-attention layer. Each of the three bars on the left represent a token embedding in $\mathcal{P}$.

Figure 5: **Gram matrix of the gradients of textual inversion token embedding across different layers.** The contribution of the $i$-th layer to the gradients of the embedding is calculated by propagating the gradients from the loss only through the $i$-th cross-attention layer. Notably, gradients coming from similar layers have remarkably higher correlation.

As $\mathcal{P}$ is a subspace of $\mathcal{P}+$, we naturally inquire about the advantages of synthesizing in the extended space. In Section 4, we present an analysis of the properties of the new space, which showcases a higher degree of control over various attributes. Specifically, different layers are found to dominate different attributes, such as style, color, and structure. We continue analysis in the supplementary in Section A.1.

A notable benefit of this space is its potential for enhancing textual inversion. We next demonstrate how the extended space can be utilized to represent subjects with greater fidelity while maintaining the capability for editing.

### 3.2 EXTENDED TEXTUAL INVERSION (XTI)

Given a set of images $\mathcal{I} = \{I_1, \ldots, I_k\}$ of a specific concept, the goal of the Textual Inversion (TI) operation Gal et al. (2022) is to find a representation of the concept in the conditioning space $\mathcal{P}$. They add a new textual token, associated with the concept, to the tokenizer model (see Figure 4, left part). This new token corresponds to an optimizable token embedding $e \in \mathcal{P}$ processed by the textual encoder. This embedding is optimized with respect to the standard diffusion denoising loss $\mathcal{L}_{\text{TI}}$ for images sampled from $\mathcal{I}$ and then used to reproduce the concept.

To motivate the need for an extended space, we start with the following experiment. Given the newly added embedding $e$, we calculate the contribution of each of the cross-attention layers to the gradients of the learned embedding. The gradients contributed by the $i$-th layer are calculated via $g_i = \frac{\partial \mathcal{L}_{\text{TI}}}{\partial e}$ where the backpropagation of the gradients from the loss is done only through the $i$-th cross-attention layer. Figure 5 depicts the expected dot product between normalized gradients $\mathbb{E}\langle \frac{g_i}{\|g_i\|}, \frac{g_j}{\|g_j\|} \rangle$ of every two cross attention layers $i, j$, averaged over different noises and images. It can be seen that gradients propagated from different cross-attention layers have lower correlation compared to gradients propagated from the same layers. Moreover, different layers may produce gradients that opposite to each other. This observation stimulated us to optimize a distinct embedding for each layer, enabling the utilization of the varied contributions of the different layers to the synthesis process.

We next explain how we extend the Extended Textual Inversion (XTI) is performed.

First, we add $n$ new textual tokens $\mathtt{t}_1, \ldots, \mathtt{t}_n$ to the tokenizer model, associated with $n$ new tokens embeddings lookup-table elements $e_1, \ldots, e_n$. Then, similarly to Gal et al. (2022), we optimize the token embeddings with the objective to predict the noise of noisy images from $\mathcal{I}$, while the token embeddings are injected into the network, one token per layer.

In practice, we employ a collection of placeholder sentences denoted by $\Pi = \{P_1, \ldots, P_m\}$, each containing a special placeholder symbol "{}" to represent the location where the tokens $\mathtt{t}_1, \ldots, \mathtt{t}_n$ is inserted (e.g. "A photograph of {}") We denote by $P_i(\mathtt{t}_1, \ldots, \mathtt{t}_n)$ the set of $n$ sentences, where the special symbol "{}" is substituted with the tokens $\mathtt{t}_1, \ldots, \mathtt{t}_n$. Assuming that the denoising U-net is parameterized by a set of parameters denoted by $\theta$, and operates within the extended conditioning space as previously described, we define the reconstruction objective for the embeddings $e_1, \ldots, e_n$ that correspond to the tokens $\mathtt{t}_1, \ldots, \mathtt{t}_n$ as follows:

$$\mathcal{L}_{\mathrm{XTI}} = \mathop{\mathbb{E}}_{\substack{P \sim \Pi, \; I \sim \mathcal{I}, \\ \varepsilon \sim \mathcal{N}(0,1), \; t}} \| \varepsilon - \varepsilon_\theta(I_t | t, P(\mathtt{t}_1, \ldots, \mathtt{t}_n)) \|_2^2$$

where $I_t$ is the image $I$ noised with the additive noise $\varepsilon$ according to the noise level $t$, and $\varepsilon_\theta$ is the noise predicted by the model. Once we operate with a latent diffusion model, we always suppose that $I$ is a latent image representation. The new look-up table embeddings $e_1, \ldots, e_n$ that correspond to $\mathtt{t}_1, \ldots, \mathtt{t}_n$ are optimized w.r.t. $\mathcal{L}_{\mathrm{XTI}}$.

# 4 EXPERIMENTS AND EVALUATION

In this section we present a comprehensive evaluation of our proposed XTI approach for the personalization task, encompassing quantitative, qualitative, and user study analysis. For more details about the user study setting please refer to the supplementary material. In supplementary Section A.1 we conduct an in-depth analysis of the various properties exhibited by the U-net cross-attention layers, and investigate how these characteristics are distributed across the layers. This analysis provides motivation for the effectiveness of our $\mathcal{P}+$ space.

In all of our experiments we use the Stable Diffusion 1.4 model Rombach et al. (2022), a latent diffusion model whose denoising U-net operates on an autoencoded image latent space. It is built on top of CLIP Radford et al. (2021), whose token embedding is represented by a vector with 768 entries, such that $\mathcal{P} \subseteq \mathbb{R}^{768}$. The U-net has four spatial resolution levels - 8x8, 16x16, 32x32, and 64x64. The 16, 32, and 64 resolution levels each have two cross-attention layers on the downward (contracting) path and three cross-attention layers on the upward (expansive) path. Resolution 8 has only 1 cross-attention layer. Thus there are a total of 16 cross-attention layers and 16 conditional token embeddings that comprise our $\mathcal{P}+ \subseteq \mathbb{R}^{768 \times 16}$ space.

## 4.1 XTI EVALUATION

We evaluate our proposed XTI and compare our results to the original Textual Inversion (TI) Gal et al. (2022). We use a combined dataset of the TI dataset of 9 concepts, and the dataset from Kumari et al. (2023) with 6 concepts. For both datasets, each concept has 4-6 original images.

We focus on TI as a baseline because it is a model-preserving inversion approach that does not fine-tune the model weights. These fine-tuning approaches like DreamBooth Ruiz et al. (2022) and Custom Diffusion Kumari et al. (2023) explicitly embed the concept within the model's output domain and thus have excellent reconstruction. However, they have several disadvantages. Firstly, they risk destroying the model's existing prior (catastrophic forgetting). Secondly, they have several orders of magnitude more parameters. Recent work with Low-Rank Adaptation (LoRA) Hu et al. (2021); Ryu (2022) reduces the number of fine-tuned parameters to a fraction, but this is still about $\sim 100$x more than XTI. Lastly, they are difficult to scale to multiple concepts since the fine-tuned parameters for each concept have to be merged. Nevertheless, we show DreamBooth as an alternative baseline for quantitative metrics.

We followed the batch size of 8 and performed 5000 optimization steps for Textual Inversion, consistent with the original paper. However, we opted to use a reduced learning rate of 0.005 without scaling for optimization, as opposed to the Latent Diffusion Model from Rombach et al. (2022) used in the original paper. In our experiments, Stable Diffusion with this learning rate worked better. For our proposed XTI, we used the same hyperparameters as for Textual Inversion, except for the number of optimization steps which we reduced to 500, **resulting in significantly faster optimization time**. Both Textual Inversion and XTI shared all other hyperparameters. On 2×Nvidia A100 GPUs, the whole optimization takes ∼15 minutes for XTI compared to ∼80 minutes for TI.

### 4.1.1 QUANTITATIVE EVALUATION

Following Gal et al. (2022), to evaluate the editability quality of the inversions, we use the average cosine similarity between CLIP embeddings of the generated images and the prompts used to generate the images (*Text Similarity*). To measure the distortion of the generated images from the original concept (*Subject Similarity*), we use the average pairwise cosine similarity between ViT-S/16 DINO Caron et al. (2021) embeddings of the generated images and the original dataset images. Compared to CLIP which is trained with supervised class labels, Ruiz et al. (2022) argued that DINO embeddings better capture differences between images of the same class due to its self-supervised training. All the methods reported in Figure 6 are evaluated over 15 subjects from Gal et al. (2022) and Kumari et al. (2023), each generated with 14 different prompts templates that place the concept in a novel context (e.g. "A photograph of {} in the jungles", see Section A.7.3 in the supplementary for details). For each test concept and prompt we generated 32 images, making a total of $15 \times 14 \times 32 = 6720$ images. We fix the generation seed across different methods.

In Figure 6 we report the evaluation of the proposed Extended Textual Inversion (XTI). Among Textual Inversion Gal et al. (2022), as for comparison we also include Dream-Booth Ruiz et al. (2022) which is not a model-preserving method. Notably, XTI outperforms TI at both subject and text similarity despite using 10x fewer training steps. We also report TI using 500 optimization steps, which is the number of steps we use for XTI. This improves the Text Similarity because fewer optimization steps prevent the optimized token embedding from being out of distribution. However, it degrades reconstruction as measured by Subject Similarity.

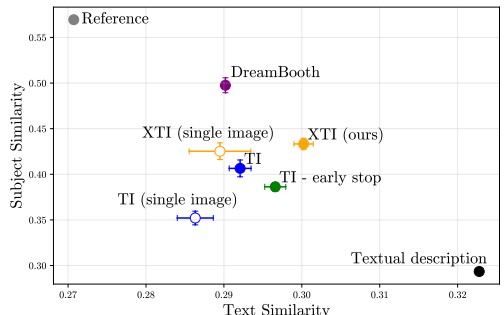

Figure 6: **Comparison of Textual Similarity and Subject Similarity.** Textual Inversion (TI) Gal et al. (2022), Extended Textual Inversion (XTI), Dream-Booth Ruiz et al. (2022). We also evaluate the metrics for both multi-image and single-image inversion setups. For the latter, a subject is represented by a single image. The "Reference" label corresponds to images containing the subject images themselves, while the "Textual description" label uses the given text description but replaces the explicit subject's description (e.g. "a colorful teapot"). The standard error is visualized in the bars.

We also report the subject inversion in a data-hungry setup, where it is represented with only a single image. Notably, even in this extreme setting, the proposed XTI performs better than multi-image TI in terms of subject similarity. As for single image training for all the runs we reduce the learning rate to 0.001 to better prevent overfitting. Figure 19 in supplementary provides a visual comparison of TI and XTI inversions in the single image setting. We omit single-image DreamBooth results from Figure 6 and 19 due to its comparatively poor performance, namely Text Similarity of 0.25 and Subject Similarity of 0.40. In particular, we found DreamBooth in this single-image setting to be prone to overfitting and difficult to optimize.

### 4.1.2 HUMAN EVALUATION

Figure 7 shows a visual comparison of our XTI approach with the original TI. Our method demonstrates less distortion to the original concept *and* to the target prompt.

To assess the efficacy of our proposed method from a human perspective, we conducted a user study. The study, summarized in Table 1, asked participants to evaluate both Textual Inversion (TI) and Extended Textual Inversion (XTI) based on their fidelity to the original subject and the given prompt. The results show a clear preference for XTI for both subject and text fidelity. Further qualitative results are presented in supplementary, Section A.2

Table 1: User study preferences for subject and text fidelity for TI and XTI. See supplementary material for more details.

| Method | Subject Fidelity | Text Fidelity |
|---|---|---|
| Textual Inversion | 24% | 27% |
| XTI (Ours) | **76%** | **73%** |

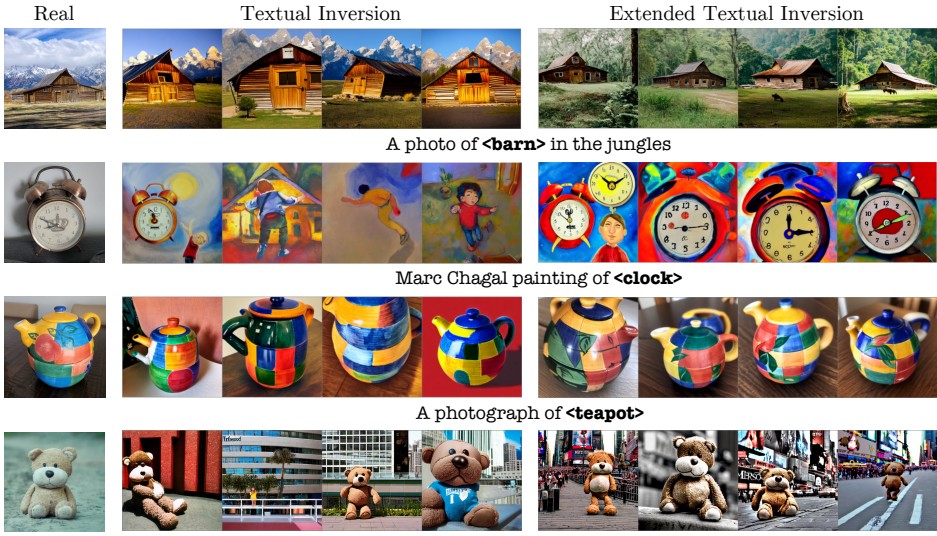

**Figure 7: Textual Inversion (TI) vs. Extended Textual Inversion (XTI).** *Column 1:* Original concepts. *Column 2:* TI results. *Column 3:* XTI results. It can be seen that XTI exhibits superior subject and prompt fidelity, as corroborated by the results of our user study.

## 4.2 EMBEDDING DENSITY

As the textual embeddings inverted with XTI have better editability properties compared to the original TI, this suggests that these tokens are better aligned with the original tokenizer look-up table embeddings, which represents the manifold of natural language embeddings. To quantify this intuition, we evaluate the density of the newly-optimized tokens with respect to the original "natural" tokens look-up table embeddings. We perform kernel-based density estimation (KDE) in the look-up table tokens embeddings space. Let us define $\mathcal{E}$ to be the set of all original tokens look-up table embeddings, before adding the extra optimized token(s).

Assuming that $\mathcal{E}$ is sampled from some continuous distribution, one can define the approximation of its density function at a point $x$ as:

$$\log p_{\mathcal{E}}(x) \approx \frac{1}{|\mathcal{E}|} \sum_{e \in \mathcal{E}} K(x - e), \quad (2)$$

where $K$ is the Gaussian kernel density function Parzen (1962); Rosenblatt (1956). As for the embeddings optimized with the original TI, this quantity always appears to be significantly smaller compared to the densities at the original embeddings $\mathcal{E}$. Figure 8 illustrates the original tokens density distribution, and the textual inversion tokens

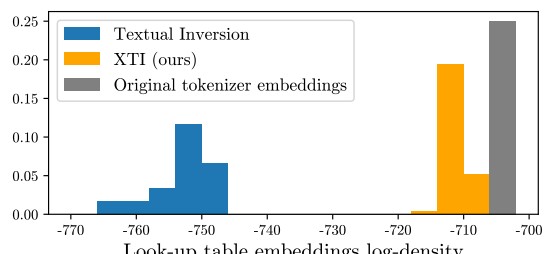

Figure 8: Estimated log-density of the original look-up table token embeddings (gray), embeddings optimized with textual inversion (blue), and embeddings optimized with XTI (orange). Our method demonstrates more regular representations which are closer to the manifold of natural words.

densities. This demonstrates that XTI provides embeddings that are closer to the original distribution, enabling more natural reconstruction and better editability.

## 5 STYLE MIXING APPLICATION

As we showed earlier, different layers of the denoising U-net are responsible for different aspects of a synthesized image. This allows us to combine the *shape* of one inverted concept with the *appearance* of another inverted concept. We call this Style Mixing.

Let us consider two independent XTI inversions of two different concepts. We can combine the inversions by passing tokens from different subjects to different layers, as illustrated in Figure 2. This mixed conditioning produces an image with a coarse geometry from the first concept and an appearance from the second concept. Formally, we are given two extended prompts: $\{p_n, \ldots, p_n\}$, and $\{q_1, \ldots, q_n\}$. We form a new extended prompt $\{p_1, \ldots, p_k, q_{k+1}, \ldots, q_K, p_{K+1}, \ldots, q_n\}$ with the separators $1 \le k < K \le n$.

Our observations indicate that the optimization of XTI with an additional density regularization loss term indicated in 2 enhances its ability to mix objects and styles, without compromising the quality of the inversion output. More details are provided in the supplementary material.

Figure 9 (right) demonstrates the combination of the `"skull mug"` and `"cat statue"` concepts from Gal et al. (2022). Different rows of the plot correspond to different blending ranges $k, K$. From top to bottom, we gradually expand it from the middle coarse layer to all the cross-attention layers. This range $(k, K)$ gives the control over the amount of details we want to bring from one inversion to another.

Figure 9 (left) shows a variety of examples generated with this method. Both shape and appearance are inherited remarkably well. For more examples and qualitative and quantitative comparisons to baselines, we refer to supplementary.

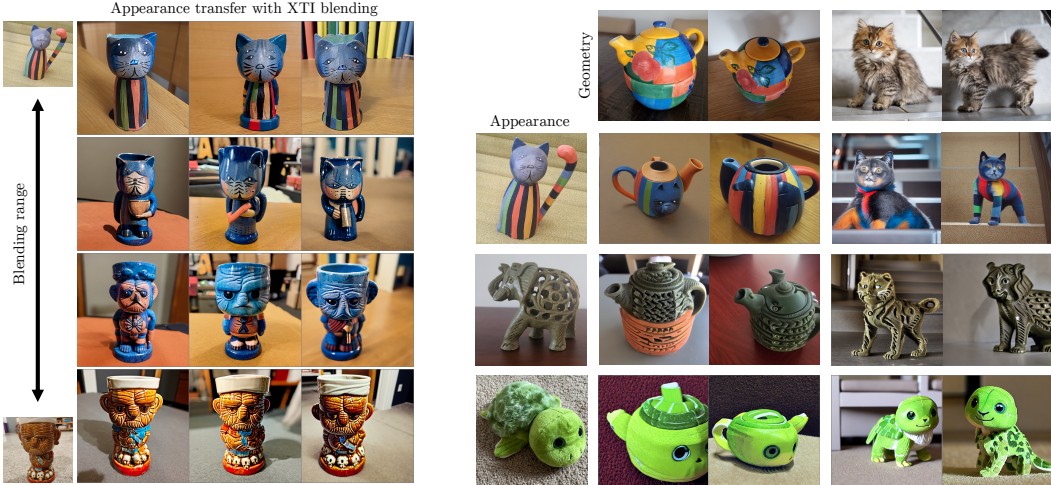

Figure 9: **Style Mixing in** $\mathcal{P}+$. *Left:* rows generated by varying the degree of mixing by adjusting the proportion of layers conditioned on either of the two $\mathcal{P}+$ inversions. *Right:* more style mixing examples. Top row shows shape source concepts, first column shows appearance source concepts.

## 6 Conclusions, Limitations, and Future work

We have presented, $\mathcal{P}+$, an extended conditional space, which provides increased expressivity and control. We have analyzed this space and showed that the denoising U-net demonstrates per-layer specification, where different layers exhibit different sensitivity to shape or appearance attributes

The competence of $\mathcal{P}+$ is demonstrated in the Textual Inversion problem. Our Extended Textual Inversion (XTI) is shown to be more accurate, more expressive, more controllable, and significantly faster. Yet surprisingly, we have not observed any reduction in editability.

The performance of XTI, although impressive, is not flawless. Firstly, it does not perfectly reconstruct the concept in the image, and in that respect, it is still inferior to the reconstruction that can be achieved by fine-tuning the model. Secondly, although XTI is significantly faster than TI, it is a rather slow process. Lastly, the disentanglement among the layers of U-net is not perfect, limiting the degree of control that can be achieved through prompt mixing.

An interesting research avenue is to develop encoders to invert one or a few images into $\mathcal{P}+$, possibly in the spirit of Gal et al. (2023), or to study the impact of applying fine-tuning in conjunction with operating in $\mathcal{P}+$.

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

## A  APPENDIX

### A.1  $\mathcal{P}+$ ANALYSIS

**Attention Analysis**  We first analyze how the distribution of the cross attention varies across layers. We create a list of 50 objects and 20 appearance adjectives (10 style descriptors and 10 texture descriptors, see Section A.7.1 for details). From these lists, we create 2000 ($= 50 \times 20 \times 2$) prompts following the patterns `"appearance object"` and `"object, appearance"`, and generate 8 images for each prompt using different seeds. We store the cross-attention values for each layer for only the object or appearance token(s), then average over the batch, spatial dimensions, and timesteps to get a ratio of attention on the object token(s) to attention on the appearance token(s).

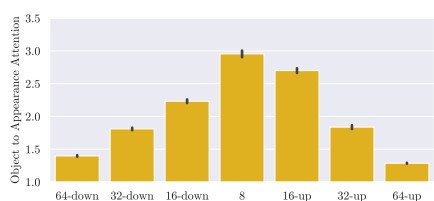

Figure 10: **Object-appearance attention ratio.** Mean ratio of attention features of the object token(s) and appearance token(s), per cross-attention layer.

Figure 10 reports the corresponding ratios. The coarse layers (8, 16) attend proportionally more to the object token and fine layers (32, 64) attend more to the appearance token. This experiment gives us the intuition that coarse layers are more responsible for object shape and structure compared to the fine layers.

Figure 11 confirms the same intuition from the self-attention layers' perspective. It shows the self-attention map, averaged over 100 randomly generated sequences and all timestamps, computed for the central latent patch. Interestingly, the lowest-resolution "bottleneck" layer performs the most widespread self-attention, while layers closer to the input and output have highly localized attention maps. This suggests that the model focuses more on global context in the low-resolution layers, while focusing more on local details in the shallow layers.

**Attributes Distribution Across Layers.**  We further analyze how different cross-attention layers impact different image attributes (shape, color, etc.). To do so, we extend the earlier `"green cube"` experiment in Figure 3 and show that this intuition can be concretely verified with the CLIP similarity metric Radford et al. (2021).

Down Layers    Up Layers

64 × 64    32 × 32    16 × 16    8 × 8    16 × 16    32 × 32    64 × 64

Figure 11: Averaged self-attention maps for the central token for different layers.

First, we divide the 16 cross-attention layers into 8 subsets, starting from the the empty set, followed by the middle coarse layer and growing outwards to include the outer fine layers, and finally the full set (see Figure 12 for a visual explanation and Section A.6 for the detailed list).

Next, we take three lists of `object`, `color`, and `style` words and randomly generate prompts with the format `"color object, style"`. For example, `"green bicycle, oil painting"` or `"red house, vector art"`. Then we randomly sample 64 pairs of these prompts. For every pair, we condition the aforementioned subset of layers on one prompt, and condition the complement set on the other prompt. We then generate 8 images with fixed seeds for each prompt-pair and subset.

As we vary the layers split ranges, we measure the similarity of the output image to each `object`, `color`, and `style` attribute of the inner (depicted blue), and outer (depicted orange) prompts with CLIP similarity. This measures the relative contribution of either conditioning prompts.

Figure 13 demonstrates this process for a single prompt pair and two image seeds. We start on the left column with all layers conditioned on the first prompt `"Blue car, impressionism"`. As we move from left to right, we depict images generated with a wider range of inner layers provided with the prompt `"Red house, graffiti"`. Note that even when some of the layers are conditioned on `"Red house, graffiti"` in the middle column, the house starts to appear red only towards the right end, when the fine layers are also conditioned on the same prompt that contains `"Red"`.

The quantitative measurements of this effect, averaged over images and prompt pairs are shown in Figure 12. We see that at either extreme, the CLIP similarities are dominated by either prompt (represented as orange or blue). However, like the example in Figure 13, different prompt attributes demonstrate different behaviors in between. We can see that it is sufficient to condition only the coarse layers for

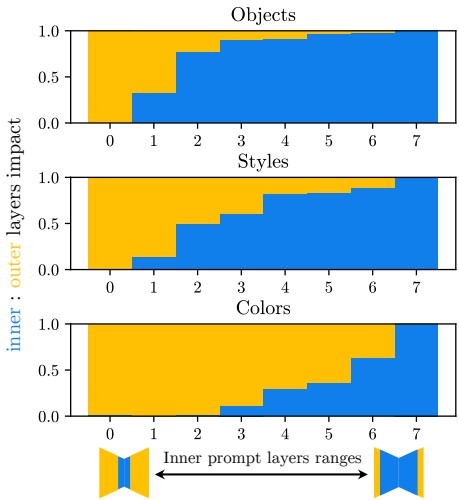

Figure 12: Relative CLIP similarities for `object`, `color` and `style` attributes, by a subset of U-net layers. Orange represents the similarity to the first prompt, and blue represents similarity to the second. As we move from left to right, we gradually grow the subset of layers conditioned with the second prompt from coarse to fine.

`object`, while `color` requires that we condition the full set of layers. `style` lies somewhere in-between. Thus, coarse layers determine the `object` shape and structure of the image, and the fine layers determine the `color` appearance of the image. `style` is a more ambiguous descriptor that involves both shape and texture appearance, so every layer has some contribution towards it.

## A.2 FURTHER MIXNIG RESULTS

Figure 14 provides a qualitative comparison between our XTI-based style mixing and baselines of TI Gal et al. (2022) and DreamBooth Ruiz et al. (2022). As for baseline we also add the MagicMix Liew et al. (2022), an approach specifically designed for appearance and object prompts mixing. There the blending is performed over different time steps in the diffusion process, while we perform blending via the different U-net layers. The results demonstrate that our approach outperforms

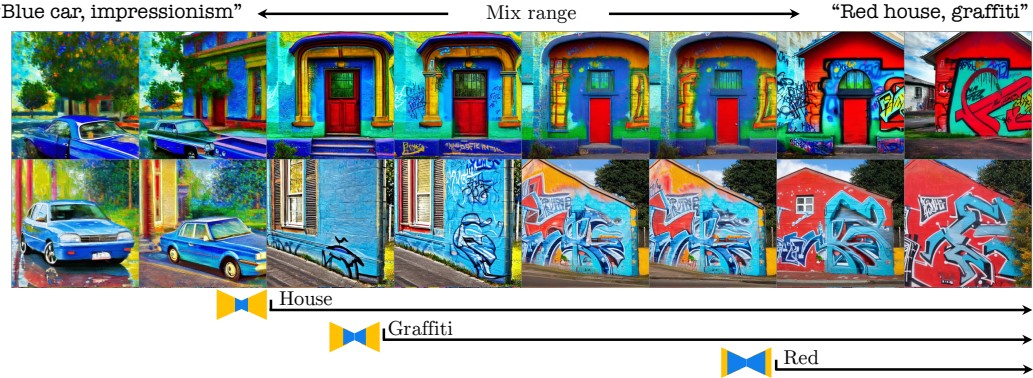

Figure 13: Visualization of mixed conditioning of the U-net cross-attention layers. The rows represent two different starting seeds and the columns represent eight growing subsets of layers, from coarse to fine. We start by conditioning all layers on `"Blue car, impressionism"` in the left column. As we move right, we gradually condition more layers on `"Red house, graffiti"`, starting with the innermost coarse layers and then the outer fine layers. Note that *shape* changes (`"house"`) take place once we condition the coarse layers, but *appearance* (`"red"`) changes only take place after we condition the fine layers.

the baselines significantly, both in terms of preserving the sources' fidelity and disentangling the attributes. In order to compare the XTI-based mixing with MagicMix we also conducted a human study asking users to pick a better appearance-geometry mixing, given geometry source sample, appearance source sample, and two sets of mixtures generated with different approaches. Among 500 questions, XTI outperforms MagicMix with the rate $81\%$ vs $19\%$.

Figure 15 shows how the mixing works with three subjects sources, showing even more fine-grained control over coarse geometry, smaller geometrical details, and coloring mixing. Figures 16 and 17 provide more examples of geometry and style mixing. In Figure 16 object prompts are passed to a wider range of layers compared to Figure 17, enforcing higher source geometry alignment.

### A.3 EXTENDED TEXTUAL INVERSION QUALITATIVE RESULTS.

Figure 18 provides more uncurated examples generated with Textual Inversion (TI) and Extended Textual Inversion (XTI). Figure 19 shows a qualitative comparison of XTI and TI optimized over a single image.

### A.4 REGULARIZATION

When applying style mixing, we discovered that optimizing XTI with an additional density regularization loss term (Equation 2) improves the mixing capability while maintaining the overall quality of the inversion. To achieve this, we use an extra loss term $-\lambda \cdot \sum_{i=1}^{n} \log p_{\mathcal{E}}(e_i)$, where $\lambda = 0.002$ serves as a small regularization scale and $e_i$ is the optimizible embedding correspondent to the $i$-th cross-attention layer. This loss term encourages the newly added look-up table embedding to be even more regular, causing the optimized tokens distribution from Figure 8 to shift closer to the original tokens distribution. We contend that this is particularly advantageous for the mixture application because in this scenario, generation is conditioned by two different XTI tokens, making it crucial to have them interact naturally, i.e., with the two tokens lying closer to the natural language manifold.

However, applying this regularization term to the original TI for subject recontextualization leads to a degradation in subject similarity of the inversion. Even with a small $\lambda$ scaling factor, this regularization enforces significant simplification in the inversion process, leaving the reconstructed token with limited freedom and expressivity. Meanwhile, although the drop in quality for XTI is minimal when the regularization is added, we do not use it by default for the recontextualization task for XTI because it would increase its complexity (another hyperparameter) and convergence rate.

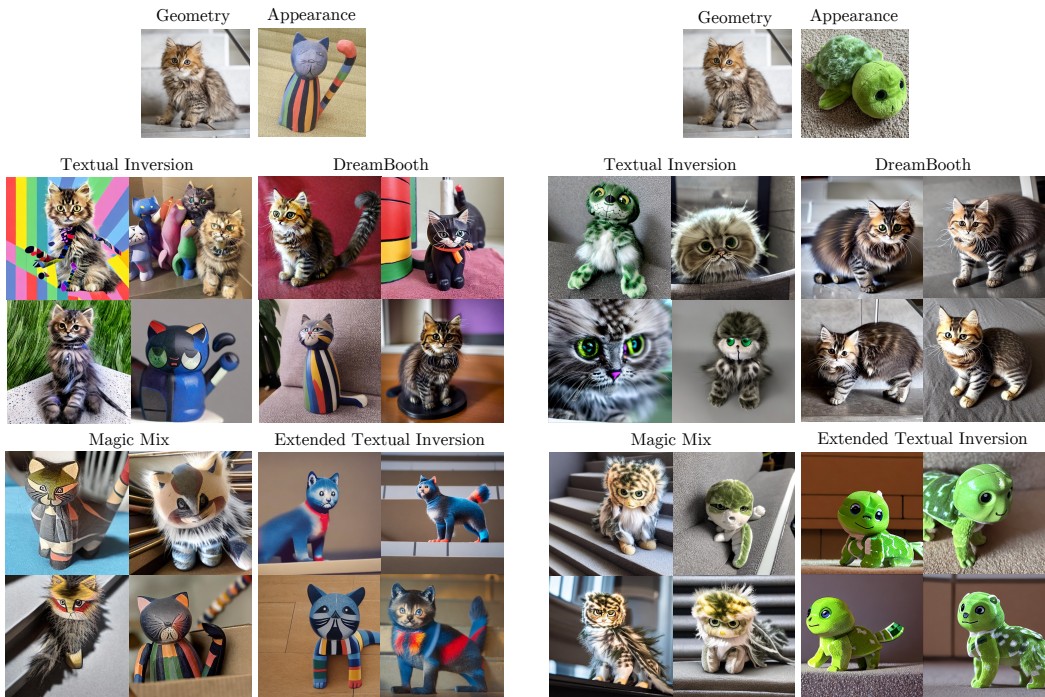

Figure 14: **Style Mixing comparison.** We compare against Textual Inversion Gal et al. (2022), Dreambooth Ruiz et al. (2022), and Magic Mix Liew et al. (2022) baselines. For TI we independently invert the target subject and the target appearance, and generate the images with the sentence "**`<object>`** `that looks like` **`<appearance>`**". Prompt variations did not make any remarkable improvements. The style source concept was inverted with style prompts (see Gal et al. (2022) for details). DreamBooth was trained with a pair of subjects. Our proposed Extended Textual Inversion clearly outperforms all three baselines.

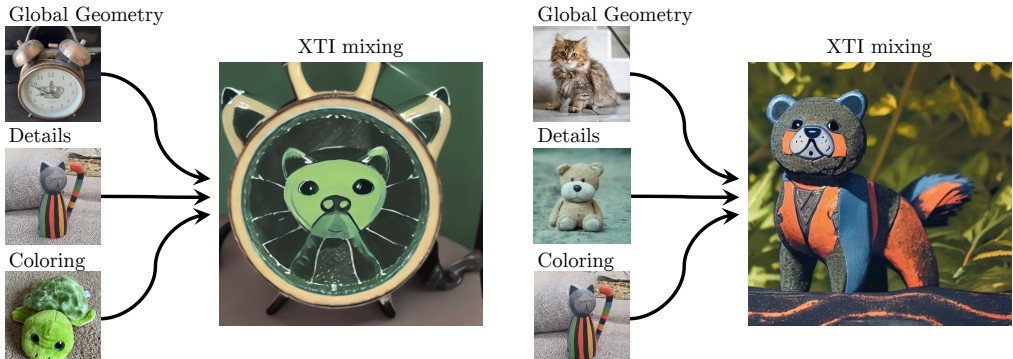

Figure 15: Examples of different subject aspects mixes for three modalities. Low-resolution layers are conditioned with a global geometry source, more shallow layers are conditioned with the middle-size details source, and high-resolution layers are conditioned with the global coloring source.

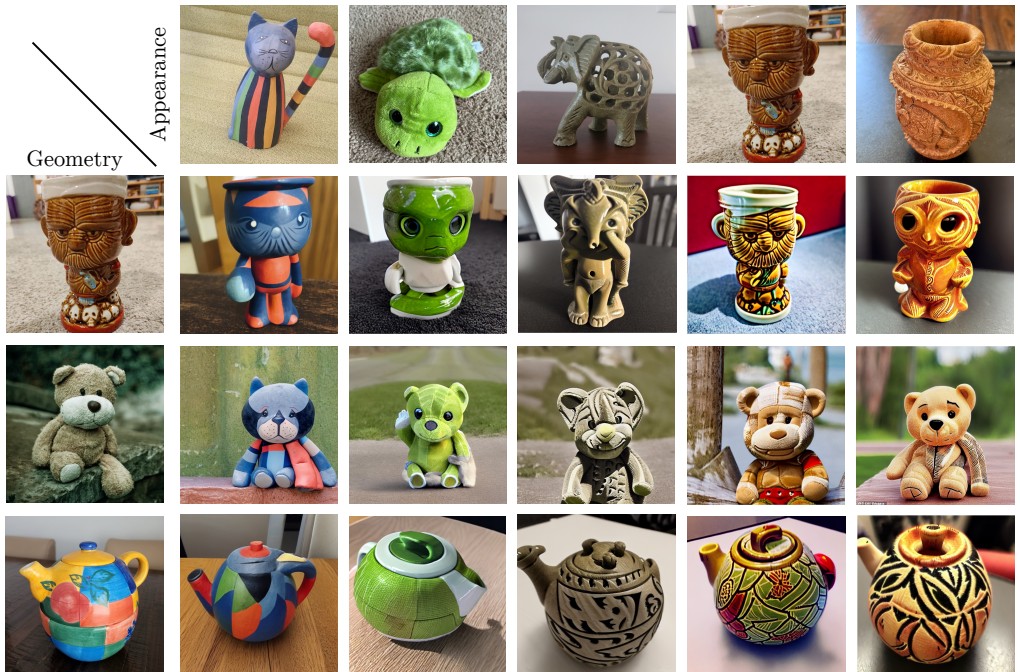

Figure 16: **Style mixing examples**. *Top row*: Style source subjects, *First column*: Geometry source subjects. The geometry subject's tokens are passed to the three layers in the range `(16, 'down', 1) – (16, 'up', 0)`, while all the rest are conditioned on the appearance subject's token.

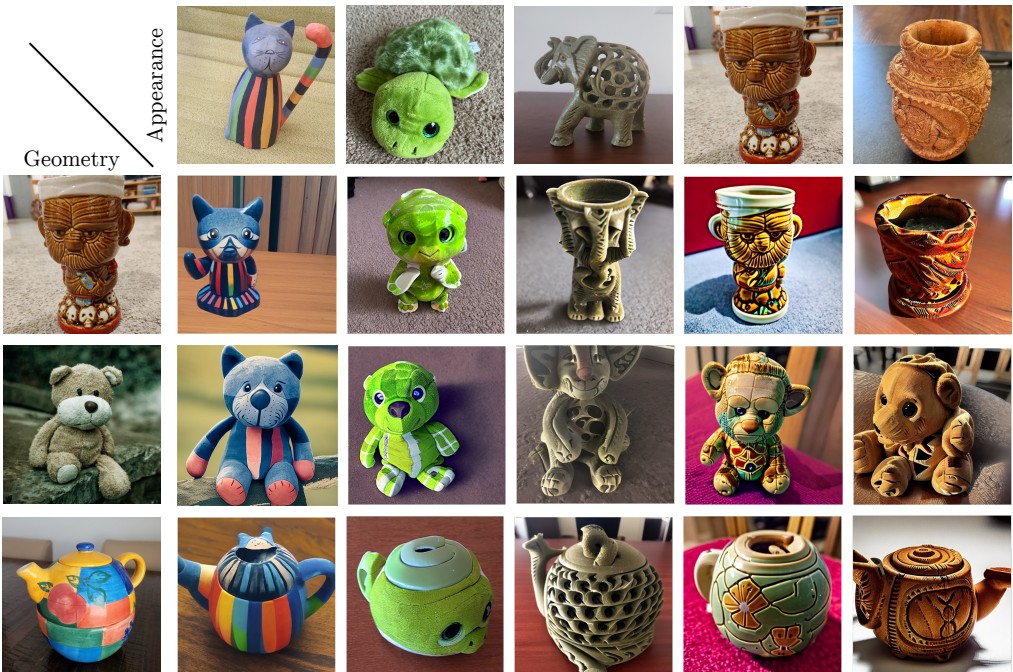

Figure 17: **Style mixing with more appearance layers**. *Top row*: Style source subjects, *First column*: Geometry source subjects. Here the geometry subject's tokens are passed only to two layers `(8, 'down', 0)` and `(16, 'up', 0)`. Thus this emphasizes the appearance subject's token more, resulting in a more dominant appearance compared to the previous setup in Figure 16.

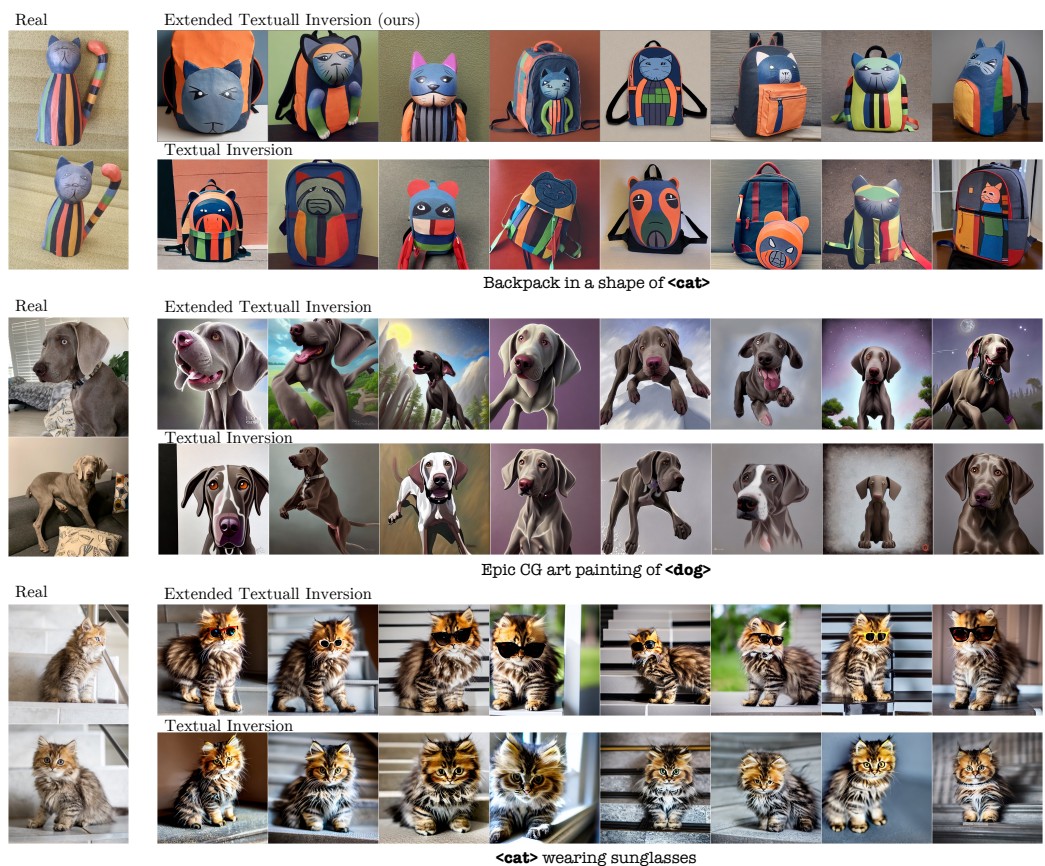

Figure 18: Samples generated with Textual Inversion (TI) and the proposed Extended Textual Inversion (XTI). XTI has better text alignment while providing more accurate subject reconstruction.

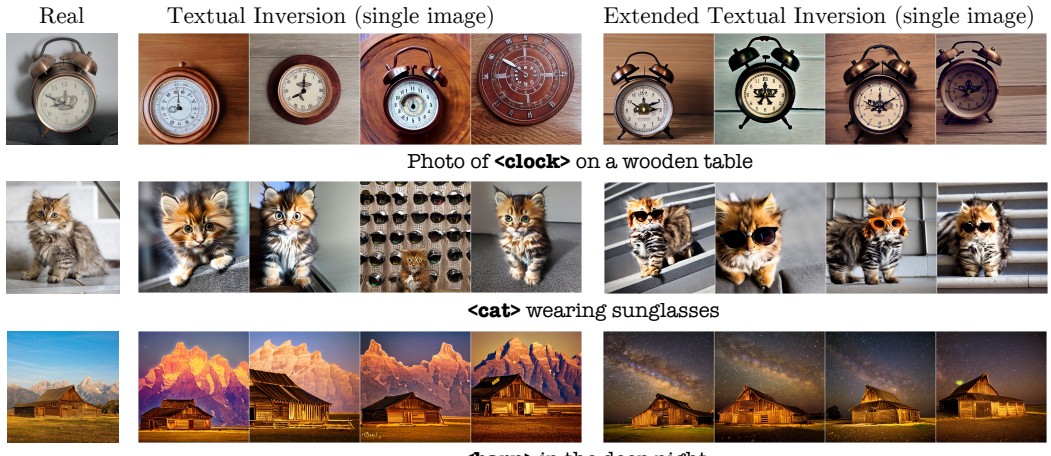

Figure 19: **Single Image Textual Inversion (TI) vs Single Image Extended Textual Inversion (XTI).** *Column 1*: Original concepts. *Column 2*: TI results. *Column 3*: XTI results. It can be seen that XTI exhibits superior subject and prompt fidelity and produces plausible results even when trained on a single image.

## A.5 EXPERIMENTS DETAILS.

As for inference we use the PNDM scheduler Liu et al. (2022) with 50 denoising steps and a classifier-free guidance Ho & Salimans (2021) scale of 7.5.

We implemented DreamBooth with a learning rate of 5e-6, a batch size of 4, and conducted 400 finetuning steps per-subject, using Stable Diffusion. Our optimization focused solely on the U-net weights and did not involve prior preservation.

For Figure 5 we used the same tokens intialization as for the inversion, the gradients are calculated for the initialized embeddings, with no optimization performed. The expectation is estimated over the same images as were used in inversion experiments, and for each of the subjects we pick 200 random noises, at different noise levels.

## A.6 COARSE/FINE LAYERS SPLIT

We provide the details of the layer subsets we used in the shape-style mixing experiments in Figures 10 and 12. We name the cross-attention layers of Stable Diffusion U-net as follows, in the order they appear in the U-net:
```
(64, 'down', 0), (64, 'down', 1), (32, 'down', 0),
(32, 'down', 1), (16, 'down', 0), (16, 'down', 1), (8, 'down', 0),
(16, 'up', 0), (16, 'up', 1), (16, 'up', 2), (32, 'up', 0),
(32, 'up', 1), (32, 'up', 2), (64, 'up', 0), (64, 'up', 1),
(64, 'up', 2).
```

The first number represents the spatial resolution, 'down' / 'up' represents whether the layer is on the downward (contracting) or upward (expansive) part of the U-net, and the third number indicates the index among the cross-attention layers of the same resolution and direction.

In Figure 12 we use the following growing sequence of cross-attention layer subsets:
0: Empty set
1: Layer (8, 'down', 0) only
2: (16, 'down', 1) – (8, 'down', 0)
3: (16, 'down', 1) – (16, 'up', 0)
4: (16, 'down', 0) – (16, 'up', 0)
5: (16, 'down', 0) – (16, 'up', 1)
6: (16, 'down', 0) – (16, 'up', 2)
7: (64, 'down', 0) – (64, 'up', 2)
The ranges listed above are inclusive.

In Figures 2 and 9 we condition the layers (8, 'down', 0), (16, 'up', 0) on the target shape textual embeddings, and the other layers on the target style textual embeddings. In Figure 3 we provide the target shape textual embeddings to layers (16, 'down', 1) – (16, 'up', 0).

## A.7 TEXT PROMPTS

### A.7.1 CROSS-ATTENTION ANALYSIS (FIGURE 10)

We used the following lists of objects and appearances for generating the "appearance object" and "object, appearance" prompts in Figure 5:

Objects (50): "dog", "cat", "tree", "chair", "book", "phone", "car", "bike", "lamp", "table", "flower", "desk", "computer", "pen", "pencil", "lamp", "television", "picture", "mirror", "shoe", "boot", "sandals", "house", "building", "street", "park", "river", "ocean", "lake", "mountain", "chair", "couch", "armchair", "bookcase", "rug", "lampshade", "fan", "conditioner", "heater", "door", "window", "bed", "pillow", "blanket", "curtains", "kitchen", "refrigerator", "stove", "oven", "microwave"

Appearances (20): `"fuzzy"`, `"shiny"`, `"bright"`, `"fluffy"`, `"sparkly"`, `"dull"`, `"smooth"`, `"rough"`, `"jagged"`, `"striped"`, `"painting"`, `"retro"`, `"vintage"`, `"modern"`, `"bohemian"`, `"industrial"`, `"rustic"`, `"classic"`, `"contemporary"`, `"futuristic"`

### A.7.2 Image Attributes Analysis (Figure 12)

For Figure 12, we used the following object, color and style words to generate the prompts:

Objects (13): `"chair"`, `"dog"`, `"book"`, `"elephant"`, `"guitar"`, `"pillow"`, `"rabbit"`, `"umbrella"`, `"yacht"`, `"house"`, `"cube"`, `"sphere"`, `'car'`

Colors (11): `"black"`, `"blue"`, `"brown"`, `"gray"`, `"green"`, `"orange"`, `"pink"`, `"purple"`, `"red"`, `"white"`, `"yellow"`

Style descriptions (7): `"watercolor"`, `"oil painting"`, `"vector art"`, `"pop art style"`, `"3D rendering"`, `"impressionism picture"`, `"graffiti"`

### A.7.3 Text Similarity Metric Prompts

For Text Similarity evaluation we use the following 14 prompts:

`"A photograph of <token>"`, `"A photo of <token> in the jungles"`, `"A photo of <token> on a beach"`, `"Aquarelle painting of <token>"`, `"Oil painting of <token>"`, `"Marc Chagall painting of <token>"`, `"Sketch drawing of <token>"`, `"Night photograph of <token>"`, `"Professional studio photograph of <token>"`, `"3d rendering of <token>"`, `"Fantasy CG art painting of <token>"`, `"A statue of <token>"`, `"A photograph of two <token> on a table"`, `"App icon of <token>"`.

Here `<token>` represents the placeholder for XTI inversion tokens to be replaced with the corresponding textual description from the following table:

Table 2: Detailed text descriptions for each dataset. The first 9 correspond to the datasets provided in Gal et al. (2022), and the remaining 6 correspond to the datasets provided in Kumari et al. (2023).

| Original Dataset | Text Description |
| --- | --- |
| elephant | a statue of an elephant |
| cat_statue | a statue of a cat |
| colorful_teapot | a colorful teapot |
| clock | an alarm clock |
| mug_skulls | a cup with a mummy |
| physics_mug | a black cup with math equations |
| red_teapot | a red teapot |
| round_bird | a round bird sculpture |
| thin_bird | a sculpture of a thin bird |
| barn | an old wooden barn |
| cat | a kitten |
| dog | a grey dog |
| teddybear | a teddy bear |
| tortoise_plushy | a tortoise plush |
| wooden_pot | an artistic wooden pot |

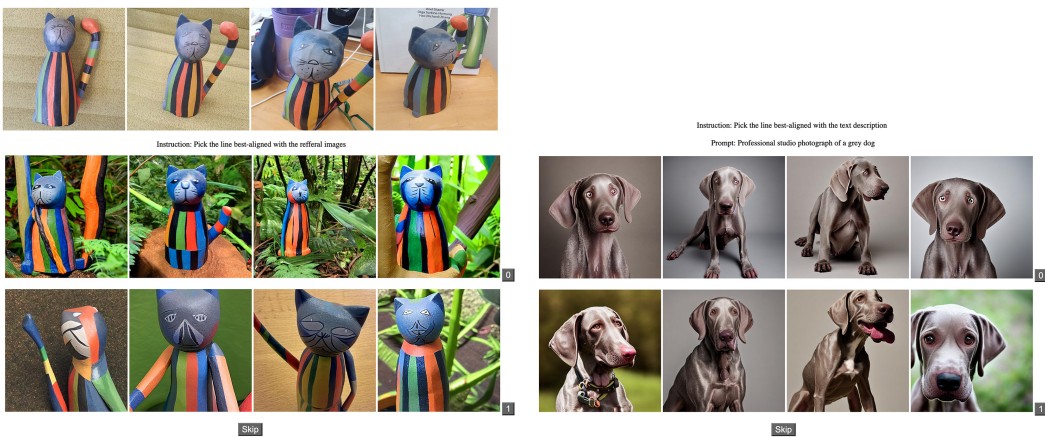

Figure 20: Human labeling interface. On the *left* we depict a sample task to evaluate subject similarity, and on the *right* the task to evaluate text similarity. The comparing methods raws are always shuffled. Both methods use the same random seed.

