# OpenReview forum: "P+: Extended Textual Conditioning in Text-to-Image Generation"
_ICLR.cc/2024/Conference — ICLR 2024 Conference Withdrawn Submission_

### Official Review · Reviewer_XoyW · 2023-10-19

**Soundness:** 3 good
**Presentation:** 3 good
**Contribution:** 2 fair
**Rating:** 5
**Confidence:** 4

**Summary:**

This paper introduces an extended text controlling space (namely P+) in T2I diffusion models, derived from per-layer prompts corresponding to a cross-attention layer of U-Net of diffusion implementation. Accordingly, the authors introduce an Extended Textual Inversion (XTI) module for converting the textual concept to this space. Experiments show the proposed method achieves better editability and reconstruction results, and allows for object-style mixing via T2I models.

**Strengths:**

- The paper is well written and easy to follow, with sufficient literature references and reasonable and intuitive design in terms of extended controlling space.

- The findings illustrated in Fig. 3 in terms of per-layer prompting is interesting.

**Weaknesses:**

- The XTI section is a bit confusing, the proposed XTI reconstruction loss (also, this equation has no numbering which makes it difficult to refer to) seems to be a stepwise loss, which means the extended space is constructed/optimized at every diffusion step of T2I models?

- Following the first point, while this operation is intuitive as many existing editing methods do follow the step-wise iterative paradigm, it is worth doing some ablations/analytical experiments on this particular operation, as recent works [a] have shown that single-step operation is sufficient to achieve semantic modification for diffusion models with more theoretical justifications.

- Following the previous two points, for the simple experiments illustrated in Fig.3, does the textual conditioning are introduced at every diffusion step?

- While the work seems to be technically solid and shows interesting results in downstream applications, most of the findings are empirical-driven, and I find it relatively difficult to interpret the proposed findings from theoretical perspectives. Do the authors have any insights on potential rationales for the layer-wise differences in U-Net on the denoising results?

[a] Boundary Guided Learning-Free Semantic Control with Diffusion Models, NeurIPS 2023.

**Questions:**

Please see the weaknesses for details.

Overall, I feel the step-wise operation is worth further investigating. To be more specific, my concern is that the statement/the proposed empirical finding “the coarse inner layers of U-Net affecting the shape of the generated image, and the outer layers affecting the style and appearance” may be inaccurate, which is more likely to be a fact of diffusion steps rather than the U-Net architecture.

---

### Official Review · Reviewer_nviY · 2023-10-27

**Soundness:** 3 good
**Presentation:** 3 good
**Contribution:** 2 fair
**Rating:** 6
**Confidence:** 3

**Summary:**

The paper introduces an advanced framework for text-to-image generation models, specifically focusing on an Extended Textual Conditioning space, referred to as P+. In contrast to the traditional methods that use a single textual condition to inform the generation process, P+ utilizes multiple textual conditions, each corresponding to a different cross-attention layer of the U-net denoising model used in the diffusion process. This extension allows for more expressive and controlled image synthesis. The authors also introduce a novel technique called Extended Textual Inversion (XTI), an advancement over the original Textual Inversion (TI) method. XTI represents a subject with a set of token embeddings.

**Strengths:**

The introduced Extended Textual Conditioning space allows for a more nuanced and controllable text-to-image synthesis.

The Extended Textual Inversion (XTI) method that improves upon the existing Textual Inversion (TI) technique is novel, and provides faster convergence and better quality.

Demonstration of groundbreaking results in object-appearance mixing through the use of the newly introduced P+ space.

**Weaknesses:**

The computational cost of XTI needs to be compared with other embedding inversion techniques. Also the inference cost compared with  standard textual conditioning, which I assume is the same?

More analysis and visualization can be done for different cross-attention layers. For example, what will happen if we provide shape textual embeddings to layer with spatial resolution of 32? I am curious of the sensitivity and affects of different layers.

**Questions:**

Have the authors also considering visualizing the distribution of different cross-attention heads? Experiments can also be done by computing contribution of each of the cross-attention heads to the gradients of the learned embedding. I wonder if there are some patterns there.

In A.6, "the third number indicates the index among the cross-attention layers of the same resolution and direction." I am curious about the meaning of the direction here.

---

### Official Review · Reviewer_UJKo · 2023-10-31

**Soundness:** 3 good
**Presentation:** 3 good
**Contribution:** 3 good
**Rating:** 5
**Confidence:** 4

**Summary:**

The paper performs Textual-Inversion in an extended space called P+, with the assumption that different cross-attention layers different artifacts of an image. Through some initial interpretability experiments, the authors show that different cross-attention layers learn distinct attributes (e.g., color or structure). With this assumption, the authors optimize layer-conditioned token-embeddings with the same objective as textual inversion.

**Strengths:**

- The observation that different cross-attention layers capture distinct attributes of an image is interesting and can be useful to understand the decision making of text-to-image diffusion models.
- Paper is extremely well written, so good job to the authors!
- The qualitative and quantitative improvements over Textual Inversion is significant (Fig. 6 and Table 1). The authors also provide a trick to reduce the optimization steps from Textual Inversion to improve the efficiency of the fine-tuning step.

**Weaknesses:**

- My primary concern with the paper is that it has not compared well with other baselines. Although other methods fine-tune some part of the diffusion model (and are expensive) — the authors should present all the results and the corresponding running time to provide the complete picture. Some of the methods which the authors should compare in completeness are: (i) Custom Diffusion (https://arxiv.org/abs/2212.04488); (ii) ELITE (https://arxiv.org/pdf/2302.13848.pdf);
- How can this method be used for multi-concept customization?
- Can the authors elaborate on what the different prompt-tokens are learning in different layers? Do the learned prompt token embeddings in the fine-layers describe the appearance? A simple experiment can be designed to check this: For e.g., obtain the token embeddings which are learned and compare it with the token embedding of the ground-truth attribute of the image (e.g., color of the object).

**Questions:**

See Weaknesses.

Overall, the paper is an interesting extension of Textual Inversion and provides some initial interpretability results for diffusion models, but lacks in discussion and comparison to existing baselines. I am happy to increase my score, if the authors can provide more discussion with existing fine-tuning based methods as described in the Weaknesses section.

---

### Official Review · Reviewer_Tj83 · 2023-11-02

**Soundness:** 3 good
**Presentation:** 3 good
**Contribution:** 2 fair
**Rating:** 5
**Confidence:** 3

**Summary:**

The paper introduces a new concept called extended textual conditioning space (P+) and a method called Extended Textual Inversion (XTI).  XTI learns unique inversed token embeddings for each cross-attention layer of the diffusion UNet. The authors claim that XTI converges faster and has novel applications compared to plain textural inversion.

**Strengths:**

- The proposed method is simple and effective.
- XTI seems effective from the experimental results under different settings. It also enables the application of style mixing.
- The writing is good and clear.

**Weaknesses:**

- The novelty can be limited. I don't see the novelty of the P+ definition or why spending paragraphs describing the P+ space is important. While the observations are interesting, I don't see novel insights introduced with P+. In fact, I think a similar observation that outer layers influence high-frequency appearance and inner layers contribute to low-frequency shape was introduced in some previous studies like Prompt-to-prompt.
- While the authors have argued about the unfair comparison between dreambooth and TI-series methods, I may not second the claim. It seems to me that XTI also introduces extra parameters than TI. Please correct me if I misunderstood this part. While XTI outperforms TI consistently, dreambooth also outperforms XTI by a large margin.
- Minor: there are quite a few related work out there for personalization, like break-a-scene, SuTI, Subject-Diffusion, and InstantBooth. I am aware that some of these methods require heavy finetuning and may not be fair baselines to consider. Yet I am not sure about the value of XTI for the community given the existence of these methods.

**Questions:**

See above.